# Boosting Generative Image Modeling via Joint Image-Feature Synthesis

**Theodoros Kouzelis**
Archimedes, Athena RC
National Technical University of Athens

**Efstathios Karypidis**
Archimedes, Athena RC
National Technical University of Athens

**Ioannis Kakogeorgiou**
Archimedes, Athena RC
IIT, NCSR "Demokritos"

**Spyros Gidaris**
valeo.ai

**Nikos Komodakis**
Archimedes, Athena RC
University of Crete
IACM-Forth

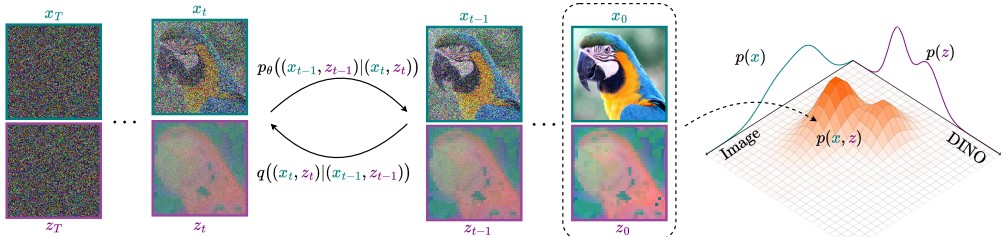

Figure 1: `ReDi`: Our generative image modeling framework bridges the gap between generative modeling and representation learning by leveraging a diffusion model that jointly captures low-level image details (via `VAE` latents) and high-level semantic features (via `DINOv2`). Trained to generate coherent image–feature pairs from pure noise, this unified latent-semantic dual-space diffusion approach significantly boosts both generative quality and training convergence speed.

## Abstract

Latent diffusion models (LDMs) dominate high-quality image generation, yet integrating representation learning with generative modeling remains a challenge. We introduce a novel generative image modeling framework that seamlessly bridges this gap by leveraging a diffusion model to jointly model low-level image latents (from a variational autoencoder) and high-level semantic features (from a pretrained self-supervised encoder like DINO). Our latent-semantic diffusion approach learns to generate coherent image–feature pairs from pure noise, significantly enhancing both generative quality and training efficiency, all while requiring only minimal modifications to standard Diffusion Transformer architectures. By eliminating the need for complex distillation objectives, our unified design simplifies training and unlocks a powerful new inference strategy: Representation Guidance, which leverages learned semantics to steer and refine image generation. Evaluated in both conditional and unconditional settings, our method delivers substantial improvements in image quality and training convergence speed, establishing a new direction for representation-aware generative modeling. Project page and code: `https://representationdiffusion.github.io/`

39th Conference on Neural Information Processing Systems (NeurIPS 2025).

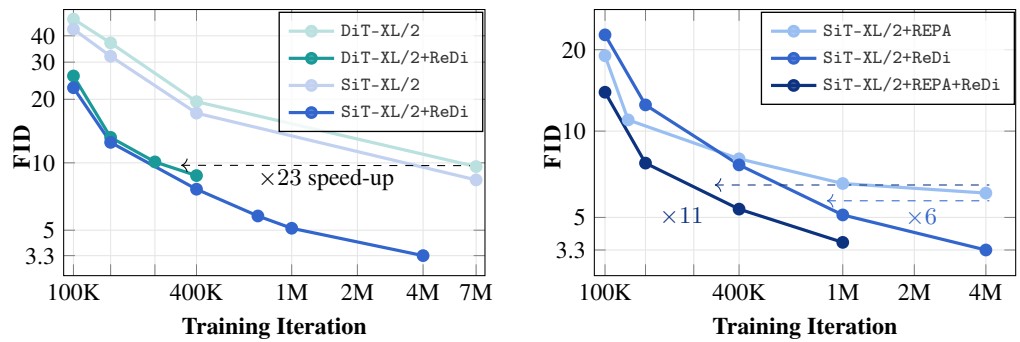

Figure 2: **Accelerated Training.** Generative performance curves on Imagenet $256 \times 256$ without Classifier-Free Guidance. **Left**: Our `ReDi` accelerates convergence of `DiT-XL/2` and `SiT-XL/2` by approximately $\times 23$. **Right:** `ReDi` converges $\times 6$ faster than `REPA`. When applied on top of `REPA` delivers a $\times 11$ speed-up.

# 1 Introduction

Latent diffusion models (`LDMs`) (Rombach et al., 2022) have emerged as a leading approach for high-quality image synthesis, achieving state-of-the-art results (Rombach et al., 2022; Peebles & Xie, 2023; Ma et al., 2024). These models operate in two stages: first, a variational autoencoder (`VAE`) compresses images into a compact latent representation (Rombach et al., 2022); second, a diffusion model learns the distribution of these latents, capturing their underlying structure.

Leveraging their intermediate features, pretrained `LDMs` have shown promise for various scene understanding tasks, including classification (Mukhopadhyay et al., 2023), pose estimation (Gong et al., 2023), and segmentation (Li et al., 2023b; Liu et al., 2023; Delatolas et al., 2025). However, their discriminative capabilities typically underperform specialized (self-supervised) representation learning approaches like masking-based (He et al., 2022), contrastive (Chen et al., 2020), self-distillation (Caron et al., 2021), or vision-language contrastive (Radford et al., 2021a) methods. This limitation stems from the inherent tension in `LDM` training - the need to maintain precise low-level reconstruction while simultaneously developing semantically meaningful representations.

This observation raises a fundamental question: *How can we leverage representation learning to enhance generative modeling?* Recent work by Yu et al. (2025) (`REPA`) demonstrates that improving the semantic quality of diffusion features through distillation of pretrained self-supervised representations leads to better generation quality and faster convergence. Their results establish a clear connection between representation learning and generative performance.

Motivated by these insights, we investigate whether a more effective approach to leveraging representation learning can further enhance image generation performance. In this work, we contend that the answer is *yes*: rather than aligning diffusion features with external representations via distillation, we propose to *jointly model both images (specifically their `VAE` latents) and their high-level semantic features* extracted from a pretrained vision encoder (e.g., `DINOv2` (Oquab et al., 2024)) within the same diffusion process. Formally, as shown in Figure 1, we define the forward diffusion process as $q(\mathbf{x}_t, \mathbf{z}_t | \mathbf{x}_{t-1}, \mathbf{z}_{t-1})$ for $t = 1, ..., T$, where $\mathbf{x}_0 = \mathbf{x}$ and $\mathbf{z}_0 = \mathbf{z}$ are the clean `VAE` latents and semantic features, respectively. The reverse process $p_\theta(\mathbf{x}_{t-1}, \mathbf{z}_{t-1} | \mathbf{x}_t, \mathbf{z}_t)$ learns to gradually denoise both modalities from Gaussian noise.

This joint modeling approach forces the diffusion model to explicitly learn the joint distribution of both precise low-level (`VAE`) and high-level semantic (`DINOv2`) features. We implement this approach, called `ReDi` (Representation Diffusion), within the `DiT` (Peebles & Xie, 2023) and `SiT` (Ma et al., 2024) frameworks with minimal modifications to their transformer architecture: we apply standard diffusion noise to both representations, combine them into a single set of tokens, and train the standard diffusion transformer architecture to denoise both components simultaneously.

Compared to `REPA`, our joint modeling approach offers three key advantages. First, the diffusion process explicitly models both low-level and semantic features, enabling direct integration of these complementary representations. Second, our method simplifies training by eliminating the need for additional distillation objectives. Finally, during inference, our unified approach enables *Representation Guidance* - where the model uses its learned semantic understanding to iteratively refine generated images, improving quality in both conditional and unconditional generation.

Our contributions can be summarized as follows:

1. We propose `ReDi`, a novel and effective method that jointly models image-compressed latents and semantically rich representations within the diffusion process, significantly improving image synthesis performance.
2. We provide a concrete implementation of our approach for both diffusion (`DiT`) and flow-matching (`SiT`) frameworks, leveraging `DINOv2` (Oquab et al., 2024) as the source of high-quality semantic representations.
3. We also introduce *Representation Guidance*, which leverages the model's semantic predictions during inference to refine outputs, further enhancing image generation quality.
4. We demonstrate that our approach boosts performance in both conditional and unconditional generation, while significantly accelerating convergence (see Figure 2).

## 2 Related work

**Representation Learning.** Various approaches aim to learn meaningful representations for downstream tasks, with self-supervised learning emerging as one of the most promising directions. Early approaches employed pretext tasks such as predicting image patch permutations (Noroozi & Favaro, 2016) or rotation angles (Gidaris et al., 2018), while more recent methods utilize contrastive learning (Chen et al., 2020; Van den Oord et al., 2018; Misra & Maaten, 2020), clustering-based objectives (Caron et al., 2020, 2018, 2019), and self-distillation techniques (Grill et al., 2020; Chen & He, 2021; Caron et al., 2021; Gidaris et al., 2021). The introduction of transformers enabled Masked Image Modeling (MIM), introduced by BEiT (Bao et al., 2022) and evolved through SimMIM (Xie et al., 2022), MAE He et al. (2022), AttMask (Kakogeorgiou et al., 2022), iBOT (Zhou et al., 2022), and MOCA (Gidaris et al., 2024), with DINOv2 (Oquab et al., 2024) achieving state-of-the-art performance through scaled models and datasets. Separately, contrastive vision-language pretraining, initiated by CLIP (Radford et al., 2021a), established powerful joint image-text representations. Subsequent models like SigLIP Zhai et al. (2023) and SigLIPv2 (Tschannen et al., 2025) refined this framework through enhanced training techniques, excelling in zero-shot settings and image retrieval (Kordopatis-Zilos et al., 2025). Building on these advances, we leverage pretrained DINOv2 visual representations to enhance image generative modeling performance.

**Diffusion Models and Representation Learning** Due to the success of diffusion models, many recent works leverage representations learned from pre-trained diffusion models for downstream tasks (Fuest et al., 2024). In particular, intermediate U-Net (Ronneberger et al., 2015) features have been shown to capture rich semantic information, enabling tasks such as semantic segmentation (Baranchuk et al., 2022; Zhao et al., 2023), semantic correspondence (Luo et al., 2023; Zhang et al., 2023; Hedlin et al., 2023), depth estimation (Zhao et al., 2023), and image editing (Tumanyan et al., 2023). Furthermore, diffusion models have been used for knowledge transfer by distilling learned representations through teacher-student frameworks (Li et al., 2023a) or refining them via reinforcement learning (Yang & Wang, 2023). Other works have shown that diffusion models learn strong discriminative features that can be leveraged for classification (Mukhopadhyay et al., 2023; Xiang et al., 2023). In a complementary direction, REPA (Yu et al., 2025) recently demonstrated that aligning the internal representations of DiT (Peebles & Xie, 2023) with a powerful pre-trained visual encoder during training significantly improves generative performance. Motivated by this observation, we propose to integrate images and semantic representations into a joint learning process.

**Multi-modal Generative Modeling** Unifying the generation across diverse modalities has recently attracted widespread interest. Notably, CoDi (Tang et al., 2023) leverages a diffusion model that enables generation across text, image, video, and audio in an aligned latent space. A joint representation for different modalities has been shown to have great scalability properties (Mizrahi et al., 2023). For video generation, WVD (Zhang et al., 2024) incorporates explicit 3D supervision by

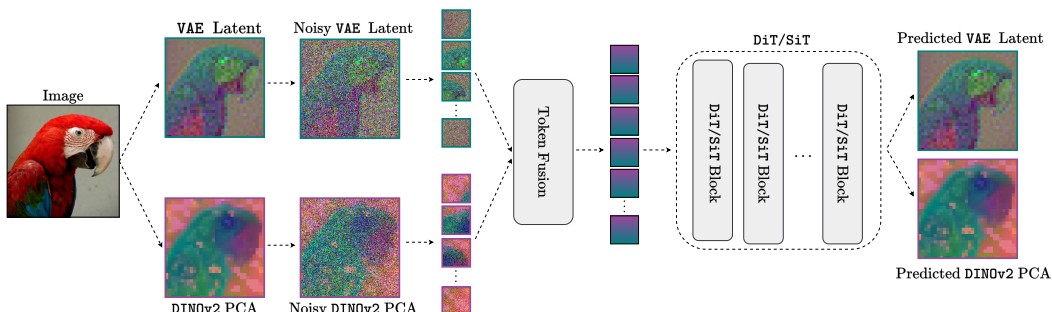

Figure 3: Given an input image, the `VAE` latent and the principal components of `DINOv2` are extracted. Both modalities are noised and fused into a *joint token sequence*, given as input to `DiT` or `SiT`.

learning the joint distribution of RGB and XYZ frames. To capture richer spatial semantics, GEM (Hassan et al., 2024) generates paired images and depth maps. MT-Diffusion (Chen et al., 2024) learns to incorporate various multi-modal data types with a multitask loss including CLIP (Radford et al., 2021b) image representations. However, they do not quantitatively assess how this impacts the generative performance. VideoJam (Chefer et al., 2025) models a joint image-motion representation that boosts temporal coherence and introduces a theoretically motivated Classifier-Free Guidance (CFG) Ho & Salimans (2022) variant to condition on both motion and text. Inspired by this approach and building on the standard CFG framework, we propose Representation Guidance, incorporating the visual representations as an additional guidance signal during inference.

## 3 Method

### 3.1 Preliminaries

**Denoising Diffusion Probabilistic Models (DDPM)**    Diffusion models (Ho et al., 2020) generate data by gradually denoising a noisy input. The forward process corrupts an input $\mathbf{x}_0$ (e.g., an image or its VAE latent) over $T$ steps by adding Gaussian noise:

$$\mathbf{x}_t = \sqrt{\bar{\alpha}_t}\mathbf{x}_0 + \sqrt{1 - \bar{\alpha}_t}\boldsymbol{\epsilon}, \tag{1}$$

where $\mathbf{x}_t$ is the noisy input at step $t$, $\bar{\alpha}_t$ are constants that define the noise schedule, and $\boldsymbol{\epsilon} \sim \mathcal{N}(\mathbf{0}, \mathbf{I})$ is the Gaussian noise term. Following Ho et al. (2020), the reverse process learns to denoise $\mathbf{x}_t$ by predicting the added noise $\boldsymbol{\epsilon}$ using a network $\boldsymbol{\epsilon}_\theta(\cdot)$ with parameters $\theta$. The training objective is:

$$\mathcal{L}_{simple} = \mathbb{E}_{\mathbf{x}_0, \boldsymbol{\epsilon}, t} \|\boldsymbol{\epsilon}_\theta(\mathbf{x}_t, t) - \boldsymbol{\epsilon}\|^2. \tag{2}$$

Although we also include the variational lower bound loss from Nichol & Dhariwal (2021) to learn the variance of the reverse process, we omit it hereafter for brevity.

Unless otherwise specified, we focus on class-conditional image generation throughout this work. For notational simplicity, we omit explicit class conditioning variables from all mathematical formulations.

**Diffusion Transformers (DiT)**    The DiT Peebles & Xie (2023) implements $\boldsymbol{\epsilon}_\theta$ using a Vision Transformer Dosovitskiy et al. (2021). Given the "patchified" input $\mathbf{x}_t \in \mathbb{R}^{L \times C_x}$ ($L$ tokens of dimension $C_x$), the model first computes embeddings:

$$\mathbf{h}_t = \mathbf{x}_t \mathbf{W}_{emb}, \quad \mathbf{W}_{emb} \in \mathbb{R}^{C_x \times C_d}. \tag{3}$$

The transformer processes $\mathbf{h}_t \in \mathbb{R}^{L \times C_d}$ to produce $\mathbf{o}_t \in \mathbb{R}^{L \times C_d}$. The final noise prediction is computed as:

$$\boldsymbol{\epsilon}_\theta(\mathbf{x}_t, t) = \mathbf{o}_t \mathbf{W}_{dec}, \quad \mathbf{W}_{dec} \in \mathbb{R}^{C_d \times C_x}. \tag{4}$$

### 3.2 Joint Image-Representation Generation

Our goal is to train a single model to jointly generate images and their semantic-aware visual representations by modeling their shared probability distribution. This approach captures the interdependent

structures and features of both modalities. While we frame our approach using DDPM, it is also applicable to models trained with flow-matching objectives Ma et al. (2024) (see Appendix A).

A high-level overview of our method is depicted in Figure 3. Let I denote a clean image, $\mathbf{x}_0 = \mathcal{E}_x(\mathbf{I}) \in \mathbb{R}^{L \times C_x}$ its VAE tokens (produced by the VAE encoder $\mathcal{E}_x(\cdot)$), and $\mathbf{z}_0 = \mathcal{E}_z(\mathbf{I}) \in \mathbb{R}^{L \times C_z}$ its patch-wise visual representation tokens (extracted by a pretrained encoder $\mathcal{E}_z(\cdot)$, e.g., DINOv2 Oquab et al. (2024))[1]. To match the spatial resolution of $\mathbf{x}_0$, we assume $\mathcal{E}_z(\cdot)$ includes a bilinear resizing operation.

During training, given $\mathbf{x}_0$ and $\mathbf{z}_0$, we define a joint forward diffusion processes:

$$\mathbf{x}_t = \sqrt{\bar{\alpha}_t}\mathbf{x}_0 + \sqrt{1 - \bar{\alpha}_t}\boldsymbol{\epsilon}_x, \quad \mathbf{z}_t = \sqrt{\bar{\alpha}_t}\mathbf{z}_0 + \sqrt{1 - \bar{\alpha}_t}\boldsymbol{\epsilon}_z, \tag{5}$$

where $\bar{\alpha}_t$ controls the noise schedule and $\boldsymbol{\epsilon}_x \sim \mathcal{N}(\mathbf{0}, \mathbf{I})$, $\boldsymbol{\epsilon}_z \sim \mathcal{N}(\mathbf{0}, \mathbf{I})$ are Gaussian noise terms of dimensions $\mathbb{R}^{L \times C_x}$ and $\mathbb{R}^{L \times C_z}$, respectively.

The diffusion model $\boldsymbol{\epsilon}_\theta(\mathbf{x}_t, \mathbf{z}_t, t)$ takes as input $\mathbf{x}_t$ and $\mathbf{z}_t$, along with timestep $t$, and jointly predicts the noise for both inputs. Specifically, it produces two separate predictions: $\boldsymbol{\epsilon}_\theta^x(\mathbf{x}_t, \mathbf{z}_t, t)$ for the image latent noise $\boldsymbol{\epsilon}_x$, and $\boldsymbol{\epsilon}_\theta^z(\mathbf{x}_t, \mathbf{z}_t, t)$ for the visual representation noise $\boldsymbol{\epsilon}_z$. The training objective combines both predictions:

$$\mathcal{L}_{joint} = \mathbb{E}_{\mathbf{x}_0, \mathbf{z}_0, t}\left[\|\boldsymbol{\epsilon}_\theta^x(\mathbf{x}_t, \mathbf{z}_t, t) - \boldsymbol{\epsilon}_x\|^2 + \lambda_z\|\boldsymbol{\epsilon}_\theta^z(\mathbf{x}_t, \mathbf{z}_t, t) - \boldsymbol{\epsilon}_z\|^2\right], \tag{6}$$

where $\lambda_z$ balances the denoising loss for $\mathbf{z_t}$. By default, we use $\lambda_z = 1$ in our experiments.

### 3.3 Fusion of Image and Representation Tokens

We explore two approaches to combine and jointly process $\mathbf{x}_t$ and $\mathbf{z}_t$ in the diffusion transformer architecture: (1) merging tokens along the embedding dimension, and (2) maintaining separate tokens for each modality (see Fig. 4). Both methods require only minimal modifications to the DiT architecture, specifically defining modality-specific embedding matrices $\mathbf{W}_{\text{emb}}^x \in \mathbb{R}^{C_x \times C_d}$ and $\mathbf{W}_{\text{emb}}^z \in \mathbb{R}^{C_z \times C_d}$, along with prediction heads $\mathbf{W}_{\text{dec}}^x \in \mathbb{R}^{C_d \times C_x}$ and $\mathbf{W}_{\text{dec}}^z \in \mathbb{R}^{C_d \times C_z}$ for $\mathbf{x}_t$ and $\mathbf{z}_t$ respectively.

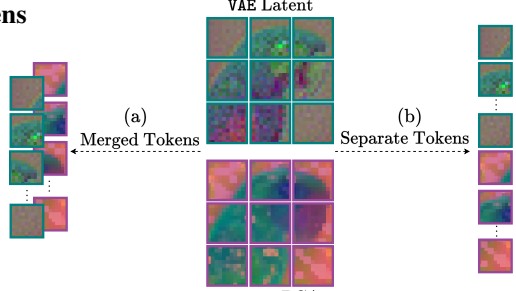

Figure 4: An illustration of our proposed token fusion approaches: (a) The tokens of the VAE latents and the DINOv2 are merged channel-wise, (b) The tokens are concatenated along the sequence dimension.

**Merged Tokens** The tokens are embedded separately and summed channel-wise:

$$\mathbf{h}_t = \mathbf{x}_t\mathbf{W}_{\text{emb}}^x + \mathbf{z}_t\mathbf{W}_{\text{emb}}^z \in \mathbb{R}^{L \times C_d}. \tag{7}$$

The transformer processes $\mathbf{h}_t$ to produce $\mathbf{o}_t$, with predictions:

$$\boldsymbol{\epsilon}_\theta^x = \mathbf{o}_t\mathbf{W}_{\text{dec}}^x, \quad \boldsymbol{\epsilon}_\theta^z = \mathbf{o}_t\mathbf{W}_{\text{dec}}^z. \tag{8}$$

This approach enables early fusion while maintaining computational efficiency, as the token count remains unchanged.

**Separate Tokens** Tokens are embedded separately and concatenated along the sequence dimension:

$$\mathbf{h}_t = [\mathbf{x}_t\mathbf{W}_{\text{emb}}^x, \mathbf{z}_t\mathbf{W}_{\text{emb}}^z] \in \mathbb{R}^{2L \times C_d}, \tag{9}$$

where $[\cdot, \cdot]$ denotes sequence-wise concatenation. The transformer outputs separate representations $\mathbf{o}_t = [\mathbf{o}_t^x, \mathbf{o}_t^z]$, with predictions:

$$\boldsymbol{\epsilon}_\theta^x = \mathbf{o}_t^x\mathbf{W}_{\text{dec}}^x, \quad \boldsymbol{\epsilon}_\theta^z = \mathbf{o}_t^z\mathbf{W}_{\text{dec}}^z. \tag{10}$$

This method provides greater expressive power by preserving modality-specific information throughout processing, at the cost of increased computation due to increased token count.

Unless stated otherwise, we use the merged tokens approach for computational efficiency.

---

[1]For notational clarity, we incorporate the patchification step (typically with $2 \times 2$ patches in DiT architectures) into the encoder definitions $\mathcal{E}_x$ and $\mathcal{E}_z$.

## 3.4 Dimensionality-Reduced Visual Representation

In practice, the channel dimension of visual representations ($C_z$) significantly exceeds that of image latents ($C_x$), i.e., $C_z \gg C_x$. We empirically observe that this imbalance degrades performance, as the model disproportionately allocates capacity to visual representations at the expense of image latents.

To address this, we apply Principal Component Analysis (PCA) to reduce the dimensionality of $\mathbf{z}_0$ from $C_z$ to $C'_z$ (where $C'_z \ll C_z$), preserving essential information while simplifying the prediction task. The PCA projection matrix is precomputed using visual representations sampled from the training set. All visual representations in Sections 3.2 and 3.3 refer to these PCA-reduced versions.

## 3.5 Representation Guidance

To ensure the generated images remain strongly influenced by the visual representations during inference, we introduce Representation Guidance. This technique during inference modifies the posterior distribution to: $\hat{p}_\theta(\mathbf{x}_t, \mathbf{z}_t) \propto p_\theta(\mathbf{x}_t)p(\mathbf{z}_t|\mathbf{x}_t)^{w_r}$, where $w_r$ controls how strongly samples are pushed toward higher likelihoods of the conditional distribution $p_\theta(\mathbf{z}_t|\mathbf{x}_t)$. Taking the log derivative yields the guided score function:

$$\nabla_{\mathbf{x}_t}\log \hat{p}_\theta(\mathbf{x}_t, \mathbf{z}_t) = \nabla_{\mathbf{x}_t}\log p_\theta(\mathbf{x}_t) + w_r\left(\nabla_{\mathbf{x}_t}\log p_\theta(\mathbf{z}_t|\mathbf{x}_t)\right) \tag{11}$$

$$= \nabla_{\mathbf{x}_t}\log p_\theta(\mathbf{x}_t) + w_r\left(\nabla_{\mathbf{x}_t}\log p_\theta(\mathbf{x}_t, \mathbf{z}_t) - \nabla_{\mathbf{x}_t}\log p_\theta(\mathbf{x}_t)\right). \tag{12}$$

By recalling the equivalence of denoisers and scores (Vincent, 2011), we implement this representation-guided prediction $\hat{\boldsymbol{\epsilon}}_{\boldsymbol{\theta}}(\mathbf{x}_t, \mathbf{z}_t, t)$ at each denoising step as follows:

$$\hat{\boldsymbol{\epsilon}}_\theta(\mathbf{x}_t, \mathbf{z}_t, t) = \boldsymbol{\epsilon}_\theta(\mathbf{x}_t, t) + w_r\left(\boldsymbol{\epsilon}_\theta(\mathbf{x}_t, \mathbf{z}_t, t) - \boldsymbol{\epsilon}_\theta(\mathbf{x}_t, t)\right). \tag{13}$$

Following Ho & Salimans (2022), we train both $\boldsymbol{e}_{\boldsymbol{\theta}}(\mathbf{x}_t, \mathbf{z}_t, t)$ and $\boldsymbol{e}_{\boldsymbol{\theta}}(\mathbf{x}_t, t)$ jointly. Specifically, during training, with probability $p_{drop}$, we zero out $\mathbf{z}_t$ (setting $\boldsymbol{\epsilon}_\theta(\mathbf{x}_t, t) = \boldsymbol{\epsilon}_\theta(\mathbf{x}_t, \mathbf{0}, t)$) and disable the visual representation denoising loss by setting $\lambda_z = 0$ in Equation 6.

# 4 Experiments

## 4.1 Setup

**Implementation details.** We follow the standard training setup of `DiT` (Peebles & Xie, 2023) and `SiT` (Ma et al., 2024), training on ImageNet at $256 \times 256$ resolution with a batch size of 256. Following ADM's preprocessing pipeline (Dhariwal & Nichol, 2021), we center-crop and resize all images to $256 \times 256$. Our experiments utilize transformer architectures `B/2`, `L/2`, and `XL/2` all using a $2 \times 2$ patch size. For unconditional generation, we simply set the number of classes to 1, maintaining the original architecture. Images are encoded into VAE latent representations using `SD-VAE-FT-EMA` (Rombach et al., 2022) that produces outputs with $\times 8$ spatial downsampling factor and 4 output channels. For $256 \times 256$ images, this results in $32 \times 32 \times 4$ latent features. Through patchification with $2 \times 2$ patches, the VAE encoder $\mathcal{E}_x(\cdot)$ yields $L = 256$ tokens, each with $C_x = 16$ channels (4 channels $\times$ $2\times2$ patch size). For semantic representation extraction, we employ `DINOv2-B` with registers (Darcet et al., 2023; Oquab et al., 2024). The 768-dimensional embeddings are reduced to 8 dimensions via PCA (trained on 76,800 randomly sampled ImageNet images). After bilinear interpolation to match the VAE's $32 \times 32 \times 4$ spatial resolution and $2 \times 2$ patchification, the encoder $\mathcal{E}_z(\cdot)$ produces $L = 256$ tokens with $C_z = 32$ channels each (8 channels $\times$ $2\times2$ patch size).

**Sampling.** For `DiT` models, we adopt DDPM sampling, while for `SiT` models, we employ the SDE Euler–Maruyama sampler. The number of sampling steps is fixed at 250 across all experiments. When using Classifier-Free Guidance (CFG) (Ho & Salimans, 2022), we apply it only to the `VAE` channels, with a guidance scale of $w = 2.4$ (see Figure 6). For Representation Guidance, we set $p_{drop} = 0.2$, the guidance scale to $w_r = 1.5$ for `B` models and $w_r = 1.1$ for `XL` models.

**Evaluation.** To benchmark generative performance, we report Frechet Inception Distance (FID) (Heusel et al., 2017), sFID (Nash et al., 2021), Inception Score (IS) (Salimans et al., 2016), Precision (Pre.) and Recall (Rec.) (Kynkäänniemi et al., 2019) using 50k samples and the ADM's TensorFlow evaluation suite (Dhariwal & Nichol, 2021).

Table 1: **FID Comparisons.** FID scores on ImageNet $256 \times 256$ without Classifier-Free Guidance for `DiT` and `SiT` models of various sizes with REPA and ReDi (ours).

| MODEL | #PARAMS | ITER. | FID↓ |
|---|---|---|---|
| DiT-L/2 | 458M | 400K | 23.2 |
| w/ REPA | 458M | 400K | 15.6 |
| w/ ReDi (ours) | 458M | 400K | 10.5 |
| SiT-L/2 | 458M | 400K | 18.5 |
| w/ REPA | 458M | 400K | 9.7 |
| w/ ReDi (ours) | 458M | 400K | 9.4 |
| DiT-XL/2 | 675M | 400K | 19.5 |
| w/ REPA | 675M | 400K | 12.3 |
| DiT-XL/2 | 675M | 7M | 9.6 |
| w/ REPA | 675M | 850K | 9.6 |
| w/ ReDi (ours) | 675M | 400K | 8.7 |
| SiT-XL/2 | 675M | 400K | 17.2 |
| w/ REPA | 675M | 400K | 7.9 |
| w/ ReDi (ours) | 675M | 400K | 7.5 |
| SiT-XL/2 | 675M | 7M | 8.3 |
| w/ REPA | 675M | 4M | 5.9 |
| w/ ReDi (ours) | 675M | 700K | 5.6 |
| w/ ReDi (ours) | 675M | 4M | 3.3 |

Table 2: **Comparison with State-of-the-art.** Quantitative evaluation on ImageNet $256 \times 256$ with Classifier-Free Guidance. Both REPA and ReDi (ours) employ `SiT-XL/2` as the base model.

| MODEL | EPOCHS | FID↓ | sFID↓ | IS↑ | PRE.↑ | REC.↑ |
|---|---|---|---|---|---|---|
| *Autoregressive Models* | | | | | | |
| VAR | 350 | 1.80 | - | 365.4 | 0.83 | 0.57 |
| MagViTv2 | 1080 | 1.78 | - | 319.4 | 0.83 | 0.57 |
| MAR | 800 | 1.55 | - | 303.7 | 0.81 | 0.62 |
| *Latent Diffusion Models* | | | | | | |
| LDM | 200 | 3.60 | - | 247.7 | 0.87 | 0.48 |
| U-ViT-H/2 | 240 | 2.29 | 5.68 | 263.9 | 0.82 | 0.57 |
| DiT-XL/2 | 1400 | 2.27 | 4.60 | 278.2 | 0.83 | 0.57 |
| MaskDiT | 1600 | 2.28 | 5.67 | 276.6 | 0.80 | 0.61 |
| SD-DiT | 480 | 3.23 | - | - | - | - |
| SiT-XL/2 | 1400 | 2.06 | 4.50 | 270.3 | 0.82 | 0.59 |
| FasterDiT | 400 | 2.03 | 4.63 | 264.0 | 0.81 | 0.60 |
| MDT | 1300 | 1.79 | 4.57 | 283.0 | 0.81 | 0.61 |
| *Leveraging Visual Representations* | | | | | | |
| REPA | 800 | 1.80 | 4.50 | 284.0 | 0.81 | 0.61 |
| ReDi (ours) | 350 | 1.72 | 4.68 | 278.7 | 0.77 | 0.63 |
| ReDi (ours) | 800 | 1.61 | 4.66 | 295.1 | 0.78 | 0.64 |

## 4.2 Enhancing the performance of generative models

**DiT & SiT.** To demonstrate the effectiveness of our approach, we present performance gains for various-sized `DiT` and `SiT` models in Table 1. Our method, `ReDi`, consistently delivers substantial improvements across models of different scales. Notably, `DiT-XL/2` with `ReDi` achieves an FID of 8.7 after just 400k iterations, outperforming the baseline `DiT-XL/2` trained for 7M steps. Similarly, `SiT-XL/2` with `ReDi` reaches an FID of 7.5 at 400k iterations, surpassing the converged `SiT-XL` at 7M steps. Additionally, Table 2 reports results for `SiT-XL/2` with Classifier-Free Guidance (CFG) Ho & Salimans (2022). Once again, `ReDi` yields significant improvements, achieving an FID of 1.72 in just 350 epochs, outperforming the baseline trained to convergence over 1400 epochs.

**Comparison with REPA.** We further compare our results with REPA, which also leverages DINOv2 features to enhance generative performance. Our approach, `ReDi`, consistently achieves superior generative performance with both `DiT` and `SiT` as the base models. As shown in Table 1, `DiT-L/2` with `ReDi` achives an FID of 10.5 significantly outperforming `DiT-L/2` with REPA. Notably, it even surpasses REPA trained for the same number of iterations with the larger `DiT-XL/2`, which achieves a higher FID of 12.3. Further for `SiT-XL` models, `ReDi` attains an FID of 5.6 in just 700k iterations, while REPA requires 4M iterations to reach an FID of 5.9. These results highlight the effectiveness of our method in leveraging visual representations to significantly boost generative performance.

**ReDi is complementary to REPA.** Interestingly, we observe that the joint modeling objective of our `ReDi` and the alignment objective of REPA are complementary. As presented in Table 5 REPA + `ReDi` matches the FID of the fully-converged REPA after only 350K iterations, and at 1M iterations reaches an FID of 3.6. For the implementation details, see Appendix B.3.

**Accelerating convergence.** The aforementioned results indicate that `ReDi` significantly accelerates the convergence of latent diffusion models. As illustrated in Figure 2, `ReDi` speeds up the convergence of `DiT-XL/2` and `SiT-XL/2` by approximately ×23, respectively. Even when compared with REPA, `ReDi` demonstrated a ×6 faster convergence. When `ReDi` is applied on top of REPA, the convergence is ×11 faster.

**Comparison with state-of-the-art generative models.** Ultimately, we provide a quantitative comparison between `ReDi` and other recent generative models using Classifier-Free Guidance (CFG)

Table 3: **Unconditional Generation FID Performance.** Results on ImageNet $256 \times 256$. For comparison, we include conditional generation results (shown in gray). Models at 400K steps. RG denotes using Representation Guidance.

| MODEL | #PARAMS | FID↓ |
|---|---|---|
| DiT-B/2 (conditional) | 130M | 43.5 |
| DiT-B/2 | 130M | 69.3 |
| w/ ReDi (ours) | 130M | 51.7 |
| w/ ReDi+RG (ours) | 130M | 47.3 |
| DiT-XL/2 (conditional) | 675M | 19.5 |
| DiT-XL/2 | 675M | 44.6 |
| w/ ReDi (ours) | 675M | 25.1 |
| w/ ReDi+RG (ours) | 675M | 22.6 |

Table 4: **FID with Representation Guidance.** FID scores on ImageNet $256 \times 256$. RG denotes Representation Guidance. Models at 400K steps.

| MODEL | #PARAMS | FID↓ |
|---|---|---|
| DiT-B/2 w/ ReDi | 130M | 25.7 |
| DiT-B/2 w/ ReDi+ RG | 130M | 20.2 |
| DiT-XL/2 w/ ReDi | 675M | 8.7 |
| DiT-XL/2 w/ ReDi+ RG | 675M | 5.9 |

Table 5: **ReDi with REPA.** FID scores on ImageNet 256×256 w/o CFG.

| MODEL | #ITER. | FID↓ |
|---|---|---|
| SiT-XL/2 w/ REPA | 4M | 5.9 |
| SiT-XL/2 w/ REPA+ReDi | 350K | 5.9 |
| SiT-XL/2 w/ REPA+ReDi | 1M | 3.5 |

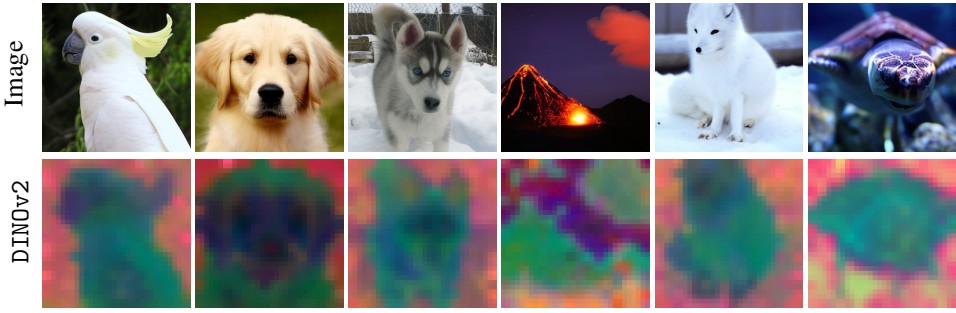

Figure 5: **Selected samples** from our SiT-XL/2 w/ ReDi model trained on ImageNet $256 \times 256$. Images and visual representations are jointly generated by our model. We use Classifier-Free Guidance with $w = 4.0$.

(Ho & Salimans, 2022) in Table 2. Our method already outperforms both the vanilla SiT-XL and SiT-XL with REPA with only 350 epochs. At 800 epochs ReDi reaches an FID of 1.64. We provide qualitative results of both generated images and visual representations in Figure 5.

**Improving Unconditional Generation.** To establish the effectiveness of our method in improving generative models, we further present experiments for unconditional generation using DiT. As shown in Table 3, our ReDi significantly improves generative performance for various model sizes. Specifically, with our ReDi FID drops from 69.3 to 51.7 for B and from 44.6 to 25.1 for XL models.

### 4.3 Impact of Representation Guidance on generative performance.

**Class Conditional Generation.** In Table 4 we present the impact of Representation Guidance (RG) on generative performance. We observe that for both B and XL models, Representation Guidance unlocks further performance enhancements by guiding the generated image to closely follow the semantic features of DINOv2. Particularly for DiT-XL w/ ReDi the FID drops from 8.7 to 5.9. We also present qualitative results in Figure 8.

**Unconditional Generation.** Representation Guidance is especially useful in unconditional generation scenarios, where the absence of class or text conditioning prevents the use of Classifier-Free Guidance to enhance performance. As demonstrated in Table 3, Representation Guidance enhances the performance of ReDi with both B and XL models, *further closing the performance gap between unconditional and conditional generation*. Notably, ReDi with Representation Guidance achieves an FID of 22.6, approaching the performance of the class-conditioned DiT-XL/2 (FID of 19.5).

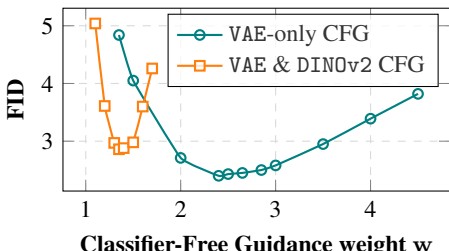

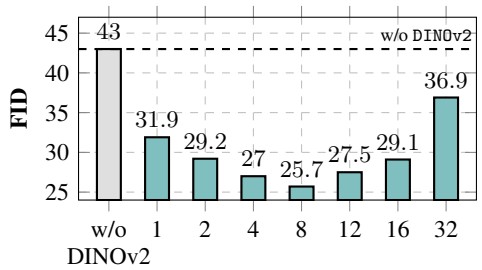

Figure 6: **VAE-only vs. VAE & DINOv2 CFG.** FID scores for SiT-XL with ReDi (trained for 400K steps) as a function of Classifier-Free Guidance weight $w$, comparing two configurations: (1) applying CFG only to VAE latents (VAE-only CFG) versus (2) applying CFG to both VAE and DINOv2 representations (VAE & DINOv2 CFG).

Figure 7: **Effect of number of principal components.** FID of DiT-B/2 w/ ReDi with different number of DINOv2 Principal Components. The vanilla DiT-B/2 is illustrated with gray. No Classifier-Free Guidance is used.

Table 6: **Performance of Modality Combination Strategies.** FID scores on ImageNet $256 \times 256$ without CFG for DiT-B/2 with ReDi using Separate Tokens (SP) and Merged Tokens (MR). See Appendix B for details on throughput measurements.

| MODEL | #TOKENS | THROUGHPUT ↑ | FID↓ |
|---|---|---|---|
| DiT-B/2 | 256 | 4.52 | 43.5 |
| w/ ReDi (MR) | 256 | 4.51 | 25.7 |
| w/ ReDi (SP) | 512 | 2.26 | 24.7 |

## 4.4 Analysis

**Dimensionality reduction ablation.** We begin the analysis of our method by ablating the impact of dimensionality reduction on the visual representations, as shown in Figure 7. Initially, we observe that jointly learning as little as one principal component yields significant improvements in generative performance. Increasing the component count continues to improve performance, up to $r = 8$, beyond which further components begin to degrade the quality of generation. This suggests an optimal intermediate subspace where compressed visual features retain sufficient expressivity to guide generation without dominating model capacity.

**Merged Tokens vs. Separate Tokens.** In Table 6, we evaluate the effectiveness of the two explored integration strategies, Merged Tokens (MR) and Separate Tokens (SP), for joint learning of image VAE latents and visual representations, using DiT-B/2 as our base model. While both approaches achieve comparable performance gains, SP demonstrates slightly better results. This advantage comes at a significant computational cost: SP doubles the transformer's input sequence length by introducing 256 additional DINOv2 tokens, resulting in approximately $2\times$ greater compute demands during both training and inference (Kaplan et al., 2020). The MR strategy, by contrast, maintains the original sequence length while delivering similar performance improvements, thereby preserving computational efficiency as measured by throughput.

**VAE-only Classifier-Free Guidance.** As ReDi jointly models both VAE latents and visual representations, we investigate two Classifier-Free Guidance (CFG) strategies: applying CFG exclusively to VAE latents (VAE-only CFG) versus applying it to both modalities simultaneously (VAE & DINOv2 CFG). Our experiments in Figure 6 demonstrate that VAE-only CFG achieves superior results, yielding an FID of 2.39 compared to 2.86 for the VAE & DINOv2 CFG approach. Notably, VAE-only CFG also shows greater robustness to variations in the CFG weight parameter.

# 5   Conclusion

In this work, we explore the relationship between semantic representation learning and generative performance in latent diffusion models. Building on recent insights, we introduced `ReDi`, a novel framework that integrates high-level semantic features with low-level latent representations within the diffusion process. Unlike prior approaches that rely on auxiliary objectives, `ReDi` jointly models the two distributions. We demonstrate that this simple approach is more effective at leveraging the semantic features and leads to drastic improvements in generative performance. We further proposed Representation Guidance, a novel guidance method that leverages the jointly learned semantic features to enhance image quality. Across both conditional and unconditional settings, `ReDi` consistently improves generation quality and accelerates convergence, highlighting the benefits of our approach.

**Acknowledgements**    This work has been partially supported by project MIS 5154714 of the National Recovery and Resilience Plan Greece 2.0 funded by the European Union under the NextGenerationEU Program and by Institute of Informatics and Telecommunications, National Center for Scientific Research "Demokritos". Hardware resources were granted with the support of GRNET. Also, this work was performed using HPC resources from GENCI-IDRIS (Grants 2024-AD011012884R3).

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

# Appendix

## Contents

## A   ReDi with Stochastic Interpolant Models (SiT)

In the main paper, we introduced `ReDi` within the DDPM framework, as employed by `DiT` models. In this section, we begin with a brief overview of Stochastic Interpolant Models Ma et al. (2024) and then describe how `ReDi` can be applied in this setting.

### A.1   Stochastic Interpolant Models (SiT)

Following flow-based models Lipman et al. (2023), stochastic interpolants involve a continuous time-dependent process transforming a data distribution $\mathbf{x_0} \sim p(\mathbf{x})$ into Gaussian noise $\boldsymbol{\epsilon} \sim \mathcal{N}(\mathbf{0}, \mathbf{I})$:

$$\mathbf{x}_t = \alpha_t \mathbf{x}_0 + \sigma_t \boldsymbol{\epsilon}, \quad \alpha_0 = \sigma_1 = 1, \quad \alpha_1 = \sigma_0 = 0, \tag{14}$$

where $\alpha_t$ and $\sigma_t$ are increasing and decreasing functions of $t$ respectively.

Given this process, the marginal probability distribution $p_t(\mathbf{x})$ of $\mathbf{x}_t$ in (14) coincides with the distribution of the probability flow ordinary differential equation with a velocity field:

$$\dot{\mathbf{x}}_t = \mathbf{v}(\mathbf{x}_t, t). \tag{15}$$

The velocity field can be approximated by a neural network $\mathbf{v}_\theta(x_t, t)$ by minimizing the following training objective:

$$\mathcal{L}_{\text{velocity}}(\theta) := \mathbb{E}_{\mathbf{x}_0, \epsilon, t} \left\| \mathbf{v}_\theta(\mathbf{x}_t, t) - \dot{\alpha}_t \mathbf{x}_0 - \dot{\sigma}_t \boldsymbol{\epsilon} \right\|^2. \tag{16}$$

### A.2   Joint Image-Representation Generation with SiT

During training, given a VAE latent image $\mathbf{x}_0$ and a visual representation $\mathbf{z_0}$, we define a joint interpolation process:

$$\mathbf{x}_t = \alpha_t \mathbf{x}_0 + \sigma_t \boldsymbol{\epsilon}_x, \quad \mathbf{z}_t = \alpha_t \mathbf{z}_0 + \sigma_t \boldsymbol{\epsilon}_z, \tag{17}$$

The model $\mathbf{v}_\theta(\mathbf{x}_t, \mathbf{z}_t, t)$ takes as input $\mathbf{x}_t$ and $\mathbf{z}_t$, along with timestep $t$, and jointly predicts the velocity for both inputs. Specifically, it produces two separate predictions: $\mathbf{v}_\theta^x(\mathbf{x}_t, \mathbf{z}_t, t)$ for the image latent velocity $\mathbf{v}_x$, and $\mathbf{v}_\theta^z(\mathbf{x}_t, \mathbf{z}_t, t)$ for the visual representation velocity $\mathbf{v}_z$. The training objective combines both predictions:

$$\mathcal{L}_{joint} = \mathbb{E}_{\mathbf{x_0}, \mathbf{z_0}, t}\left[\|\mathbf{v}_\theta^x(\mathbf{x}_t, \mathbf{z}_t, t) - \dot{\alpha}_t\,\mathbf{x}_0 - \dot{\sigma}_t\,\boldsymbol{\epsilon}_x\|^2 + \lambda_z \|\mathbf{v}_\theta^z(\mathbf{x}_t, \mathbf{z}_t, t) - \dot{\alpha}_t\,\mathbf{z}_0 - \dot{\sigma}_t\,\boldsymbol{\epsilon}_z\|^2\right], \quad (18)$$

where $\lambda_z$ balances the velocity loss for $\mathbf{z}_t$. By default, we use $\lambda_z = 1$, $\alpha_t = t$ and $\sigma_t = 1 - t$ in our experiments.

## B  Additional Implementation Details

### B.1  Architecture details

We present in Table 7 the configurations of the different-sized `DiT` and `SiT` models used in our experiments.

Table 7: **Model configuration details.** The configurations are the same for both `DiT` and `SiT` models.

| MODEL SIZE | B/2 | L/2 | XL/2 |
|---|---|---|---|
| Input Size | $32 \times 32 \times 4$ | $32 \times 32 \times 4$ | $32 \times 32 \times 4$ |
| Patch Size | 2 | 2 | 2 |
| # Layers | 12 | 24 | 28 |
| # Heads | 12 | 16 | 16 |
| Hidden Dim. | 768 | 1024 | 1152 |

### B.2  Optimization details

We present in Table 8 the optimization hyperparameters used for all experiments presented in the paper.

Table 8: **Optimization details.** The optimization hyperparameters for both `DiT` and `SiT` models.

| | |
|---|---|
| Batch Size | 256 |
| Optimizer | AdamW |
| LR | $10^{-4}$ |
| $(\beta_1, \beta_2)$ | $(0.9, 0.999)$ |

**Computational Resources.**  For both training and sampling we use 8 NVIDIA A100 40GB GPUs. Throughput, as presented in Table 6 is measured on a single NVIDIA A100 40GB GPU with a batch size of 64 as the number of images generated per second using 250 sampling steps.

### B.3  Further implementation details

**ReDi with REPA experiment.**  To apply the Representation Alignment objective (`REPA`) on top of `ReDi` we follow the implementation of (Yu et al., 2025) and employ a projection layer in the 8th transformer layer. The projection is a three-layer MLP with SiLU activations (Elfwing et al., 2018). The weight on alignment loss is $\lambda_{\texttt{REPA}} = 0.5$.

## C  Detailed Benchmarks

We provide a detailed evaluation of the main experiments presented in the main paper, including additional metrics and training iterations. Specifically, Table 9 details the performance of the `SiT-XL/2 w/ ReDi` models. Further Table 10 presents results for the `ReDi with REPA (SiT-XL/2)`. For all models, we use the evaluation metrics reported in the original publications.

| MODEL | #ITERS. | FID↓ | sFID↓ | IS↑ | PREC.↑ | REC.↑ |
|---|---|---|---|---|---|---|
| SiT-XL/2 Peebles & Xie (2023) | 7M | 8.3 | 6.3 | 131.7 | 0.68 | 0.67 |
| w/ ReDi | 50K | 56.1 | 18.9 | 23.8 | 0.44 | 0.47 |
| w/ ReDi | 100K | 23.1 | 5.9 | 61.5 | 0.64 | 0.57 |
| w/ ReDi | 200K | 12.6 | 5.7 | 97.3 | 0.69 | 0.61 |
| w/ ReDi | 300K | 9.7 | 5.3 | 117.3 | 0.71 | 0.62 |
| w/ ReDi | 400K | 7.5 | 5.1 | 129.5 | 0.72 | 0.62 |
| w/ ReDi | 4M | 3.3 | 4.8 | 188.9 | 0.74 | 0.68 |

Table 9: **Detailed evaluation** for SiT-XL/2 w/ ReDi. All results are reported without classifier-free guidance.

| MODEL | #ITERS. | FID↓ | sFID↓ | IS↑ | PREC.↑ | REC.↑ |
|---|---|---|---|---|---|---|
| SiT-REPA-XL/2 Yu et al. (2025) | 400K | 7.9 | 5.1 | 122.6 | 0.70 | 0.65 |
| SiT-REPA-XL/2 | 4M | 5.9 | 5.7 | 157.8 | 0.70 | 0.69 |
| w/ ReDi | 50K | 44.8 | 18.7 | 32.8 | 0.50 | 0.49 |
| w/ ReDi | 100K | 15.2 | 5.6 | 85.3 | 0.68 | 0.59 |
| w/ ReDi | 200K | 8.3 | 5.2 | 122.3 | 0.71 | 0.61 |
| w/ ReDi | 300K | 6.3 | 5.1 | 140.6 | 0.73 | 0.62 |
| w/ ReDi | 400K | 5.3 | 4.9 | 149.8 | 0.74 | 0.63 |
| w/ ReDi | 1M | 3.5 | 4.64 | 177.9 | 0.75 | 0.69 |

Table 10: **Detailed evaluation** for ReDi with REPA. All results are reported without classifier-free guidance.

# D  Baseline Generative Models

We provide here a brief description of the baseline approaches presented in the main paper. Specifically, we consider (a) *Autoregressive Models*, (b) *Latent Diffusion Models*, and (c) REPA (Yu et al., 2025) that also *leverages visual representations* to enhance generative performance.

## (a) Autoregressive Models

- VAR (Tian et al., 2024) proposes a scalable generative framework that autoregressively predicts higher-resolution image details from lower-resolution contexts across multiple scales.

- MagViTv2 (Yu et al., 2024) introduces a lookup-free quantization method enabling a large vocabulary that is able to improve the generation quality of autoregressive models.

- MAR (Li et al., 2024) proposes an autoregressive image generation framework that eliminates the need for vector quantization

## (b) Latent Diffusion Models

- LDM (Rombach et al., 2022) proposes latent diffusion models, modeling the image distribution in a compressed latent space produced by a KL- or VQ-regularized autoencoder.

- U-ViT-H/2 Bao et al. (2023) proposes a ViT-based (Dosovitskiy et al., 2021) latent diffusion model that incorporates skip connections.

- DiT Peebles & Xie (2023) proposes a pure transformer backbone for training diffusion models and incorporates AdaIN-zero modules.

- MaskDiT (Zheng et al., 2023) trains diffusion transformers with an auxiliary mask reconstruction task

- MDT Gao et al. (2023) introduce an effective mask latent modeling scheme, and design an asymmetric masking diffusion transformer.

- `SD-DiT` (Zhu et al., 2024) extends the MaskDiT architecture by incorporating a discrimination objective using a momentum encoder.
- `SiT` (Ma et al., 2024) improves diffusion transformer training by moving from discrete diffusion to continuous flow-based modeling.
- `FasterDiT` (Yao et al., 2024) incorporates supervision of the velocity direction into the denoising objective, significantly accelerating the training process.

**(c) Leveraging Visual Representations**

- `REPA` (Yu et al., 2025) aligns the representations of diffusion transformer models to the representations of self-supervised models.

# E    Limitations & Future Work

This section outlines some limitations of our current work and highlights promising directions for future research.

**Multiple visual representations.**    In this work, we demonstrate the effectiveness of jointly modeling the visual representations from `DINOv2` during the diffusion process. A promising direction for future research is to investigate whether integrating *multiple* visual representations, each capturing different semantic or structural properties, can further boost generative performance.

**Different dimensionality reduction approaches.**    We have shown that projecting visual representations into a lower-dimensional space with PCA effectively compresses visual features while retaining sufficient information. An interesting direction for future work is to explore more sophisticated compression techniques, such as training an autoencoder, to better capture and retain the expressivity of these features.

# F    Broader Impact

Generative models carry a substantial risk of misuse. Their application can lead to various negative societal impacts, most notably the spread of disinformation. Enhancements in generative performance, as achieved by our method, may further increase the realism of generated content, potentially making disinformation even more convincing.

# G   Additional Qualitative Results

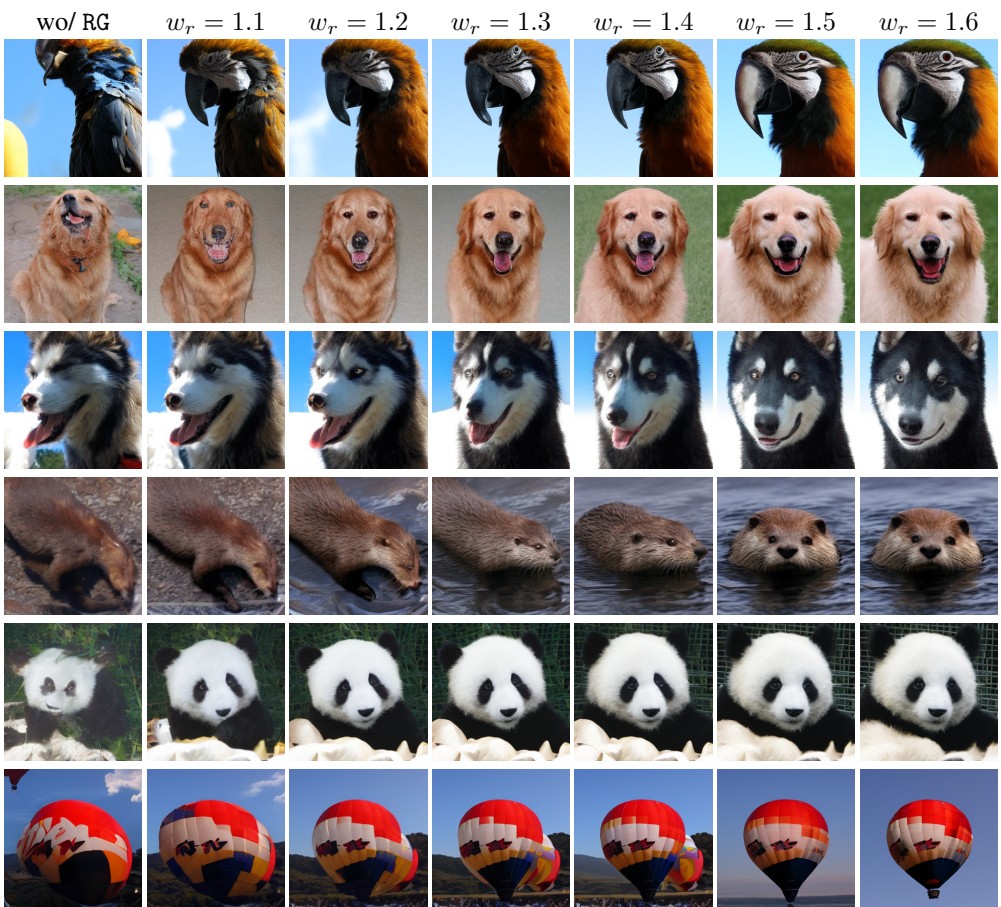

Figure 8: **The effect of Representation Guidance.** Samples from our `DiT-XL/2` w/ `ReDi` model trained on ImageNet $256 \times 256$ for 400k steps with different Representation Guidance weights $w_r$.

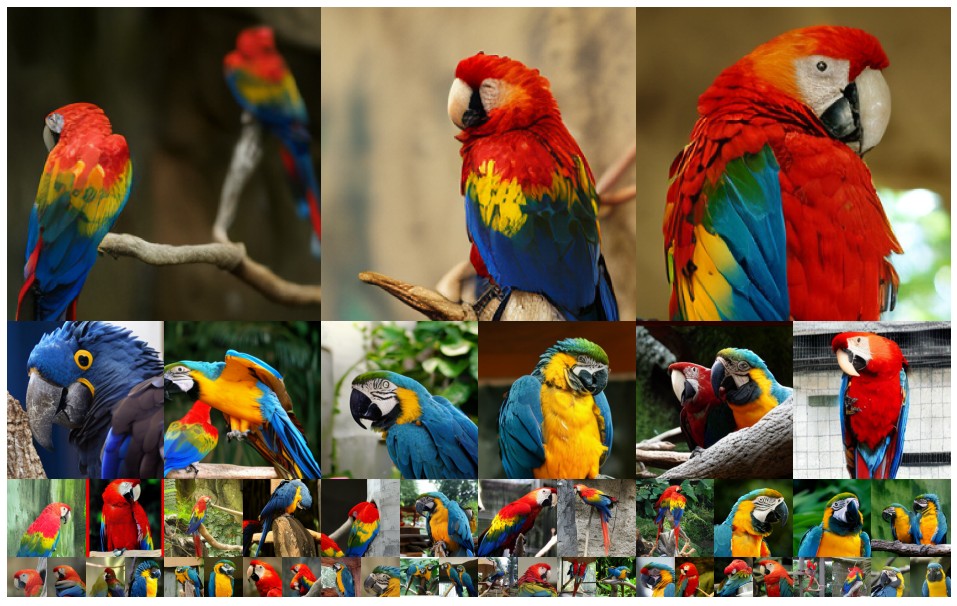

Figure 9: **Uncurated generation results** of `SiT-XL/2` w/ `ReDi`. We use Classifier-Free Guidance with $w = 4.0$. Class label = 88.

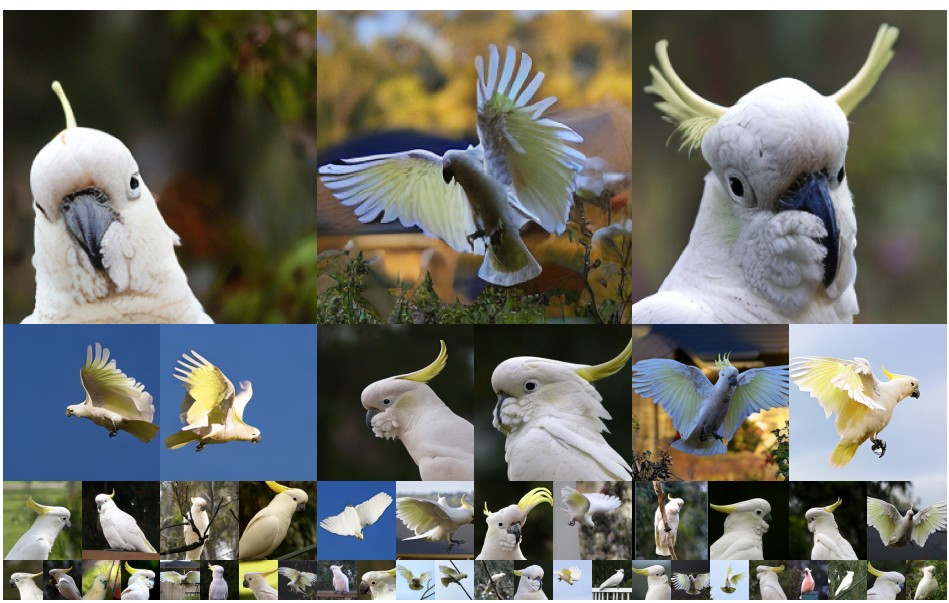

Figure 10: **Uncurated generation results** of `SiT-XL/2` w/ `ReDi`. We use Classifier-Free Guidance with $w = 4.0$. Class label = 89.

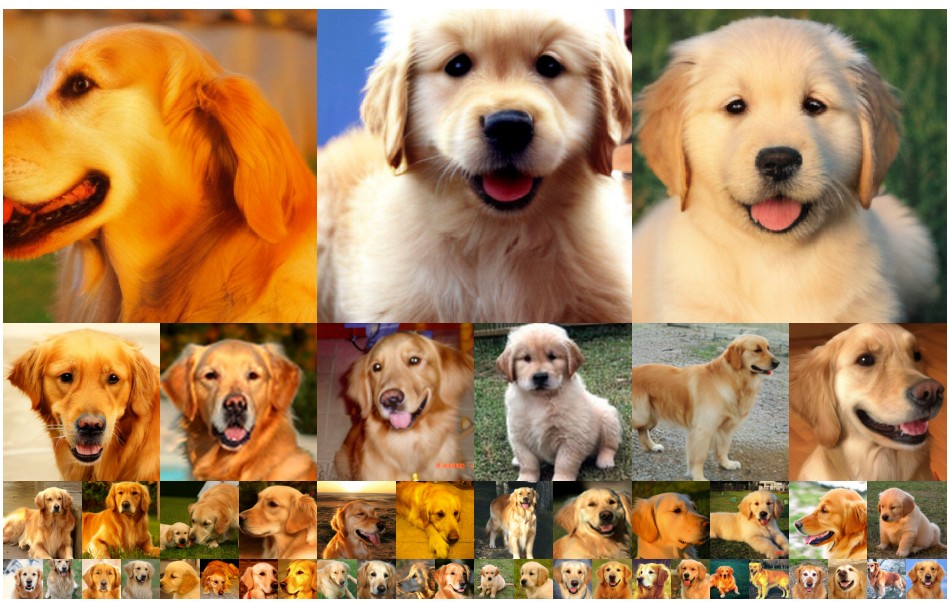

Figure 11: **Uncurated generation results** of `SiT-XL/2 w/ ReDi`. We use Classifier-Free Guidance with $w = 4.0$. Class label = 207.

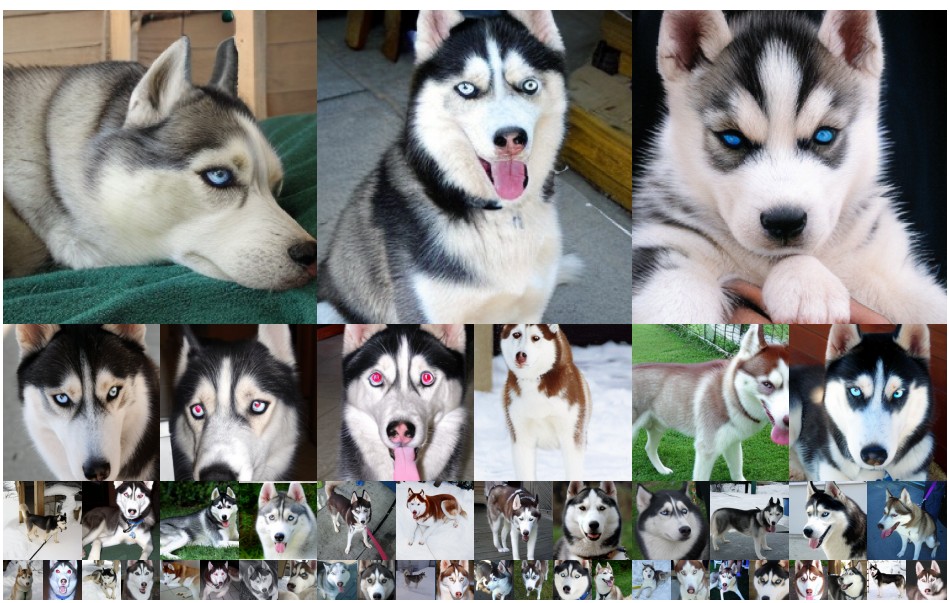

Figure 12: **Uncurated generation results** of `SiT-XL/2 w/ ReDi`. We use Classifier-Free Guidance with $w = 4.0$. Class label = 250.

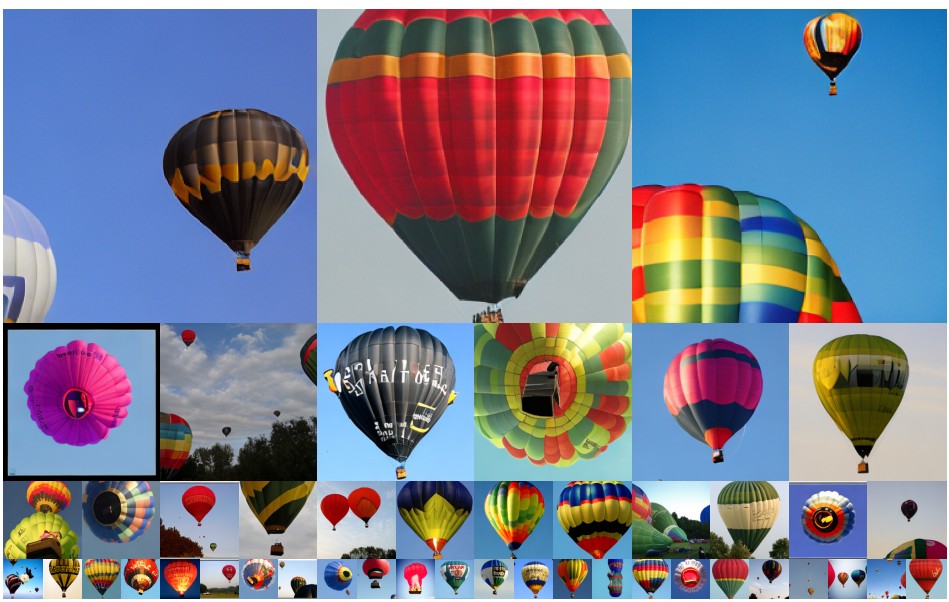

Figure 13: **Uncurated generation results** of `SiT-XL/2` w/ `ReDi`. We use Classifier-Free Guidance with $w = 4.0$. Class label = 417.

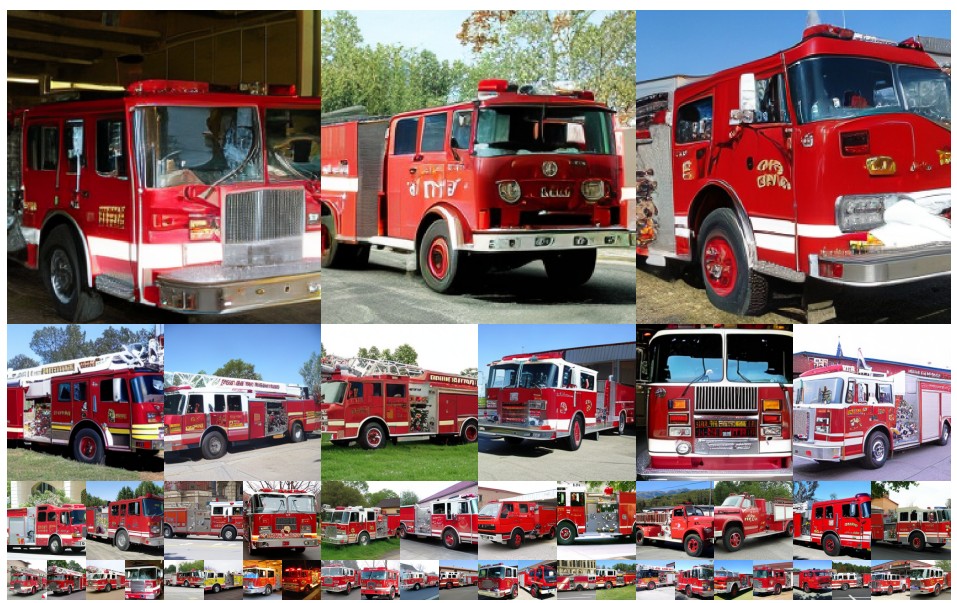

Figure 14: **Uncurated generation results** of `SiT-XL/2` w/ `ReDi`. We use Classifier-Free Guidance with $w = 4.0$. Class label = 555.

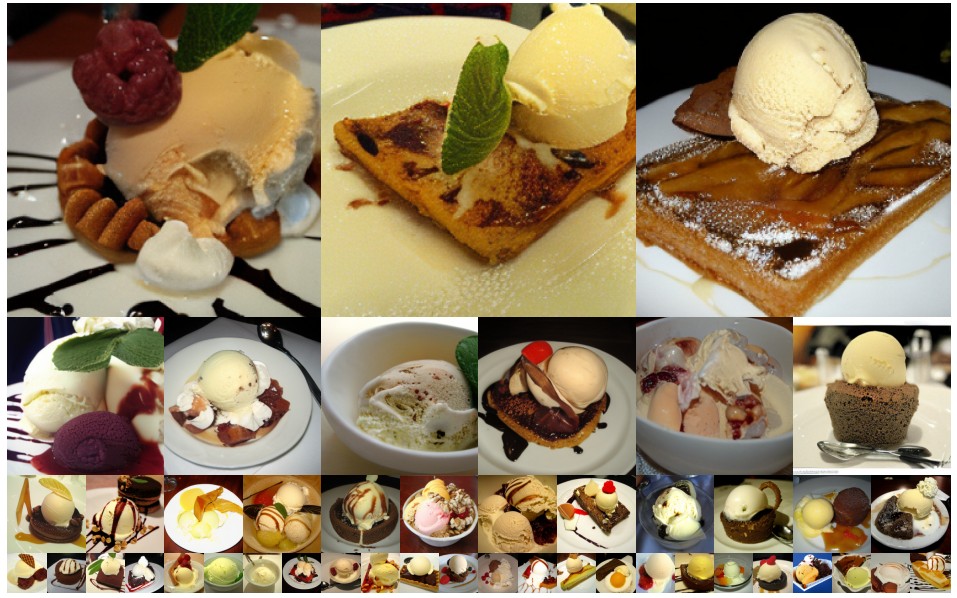

Figure 15: **Uncurated generation results** of `SiT-XL/2` w/ `ReDi`. We use Classifier-Free Guidance with $w = 4.0$. Class label = 928.

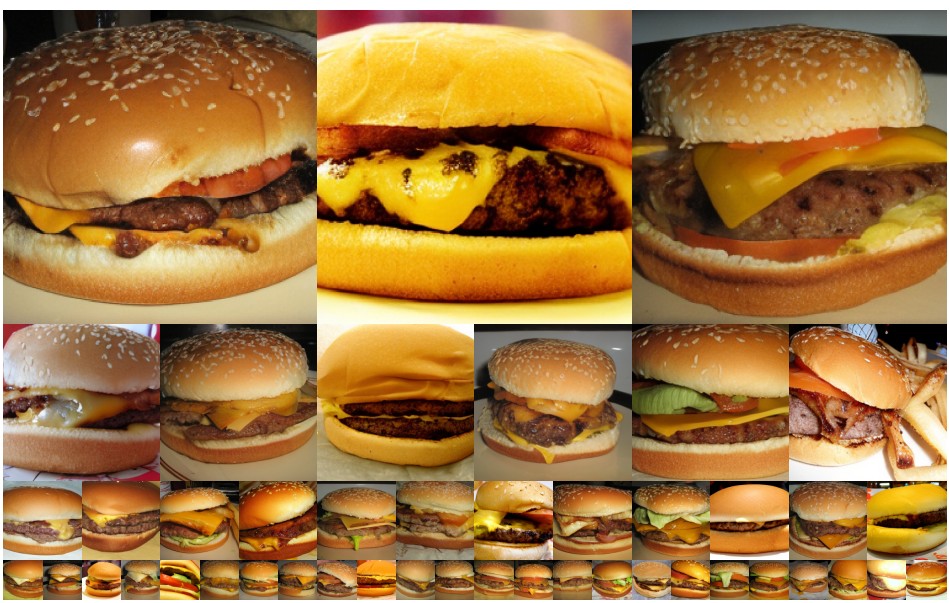

Figure 16: **Uncurated generation results** of `SiT-XL/2` w/ `ReDi`. We use Classifier-Free Guidance with $w = 4.0$. Class label = 933.

