# OpenReview forum: "Boosting Generative Image Modeling via Joint Image-Feature Synthesis"
_NeurIPS.cc/2025/Conference — NeurIPS 2025 spotlight_

### Official Review · Reviewer_8DbC · 2025-06-20

**Clarity:** 4
**Significance:** 3
**Originality:** 3
**Rating:** 5
**Confidence:** 4

**Summary:**

This paper proposes a novel generative paradigm named joint image-feature synthesis ReDi. Inspired by REPA , this paradigm first performs understanding (feature synthesis) followed by synthesis (image synthesis) within the backbone's forward pass. The significance of this method lies in two aspects: (1) Its novelty may provide valuable insights for designing future unified understanding and synthesis algorithms , and (2) it has the potential to enhance the model's generalization capabilities and training speed.

**Questions:**

No.

**Ethical Concerns:**

["NO or VERY MINOR ethics concerns only"]

**Limitations:**

Yes.

**Paper Formatting Concerns:**

No.

**Quality:**

3

**Strengths And Weaknesses:**

**Strength**:

1. The greatest strength of this paper lies in its highly interesting and novel algorithm. Additionally, its proposed representation guidance also offers valuable insights.

2. This paper is written in a very logical manner and the experiments are adequate and sound.

3. The experimental results are outstanding; ReDi outperforms REPA in both model performance and training speed.

**Weakness**:

1. The paper lacks some analytical experiments, such as: the FID change curve during training, a visualization of the "synthesized" representation's evolution during the sampling process, and a comparison of the "synthesized" representations between REPA and ReDi to better explain why ReDi performs better.

2. In addition to DINOv2, the authors should explore other representations, such as MAE, LLM Embeddings (e.g., Gemma and T5 Embedding) corresponding to prompts, or even representations from a REPA pre-trained diffusion model.

3. The statement "This suggests an optimal intermediate subspace where compressed visual features retain sufficient expressivity to guide generation without dominating model capacity" only provides an intuitive explanation of the experimental results. Is there a more in-depth explanation regarding Figure 7?

---

> ### Author Rebuttal · Authors · 2025-07-31
>
> We appreciate your insightful comments and efforts in reviewing our manuscript. Below, we provide our responses to each of your comments:
>
> **W1. Lack of some analytical experiments**
>
> - **FID change curve during training.**
>
> We respectfully emphasize that in Figure 2, we demonstrate the progression of FID during training for both DiT-XL w/ ReDi and SiT-XL w/ ReDi, and compare with the vanilla DiT/SiT and REPA.
> - **Visualization of the "synthesized" representations sampling process.**
>
>  We appreciate the reviewer’s suggestion. Because the NeurIPS 2025 rebuttal guidelines do not allow new figures to be uploaded at this stage, we cannot include the requested visualization in the rebuttal. We will, however, add a figure illustrating how ReDi’s synthesized representations evolve over the course of inference in the camera‑ready version.
> - **Comparison of the "synthesized" representations between REPA and ReDi.**
>
> We appreciate the reviewer’s suggestion to include a comparison between the features generated by ReDi and REPA. We will include a visual comparison of the generated features from both methods in the final version of our paper.
>
>
>
> **W2. Alternative Representations**
>
> We thank the reviewer for their suggestion to explore alternative representations.
> Although we selected DINOv2 for its state‑of‑the‑art performance across a wide spectrum of vision tasks, we agree that evaluating other representations can provide additional insight.
>
> To address this concern, we conducted some preliminary supplementary experiments using 2 alternative representations, namely MAE and JEPA, with different number of principal components (#PC). The additional experiments suggested by the reviewer, such as LLM embeddings corresponding to prompts and representations from a REPA, are not trivial to implement within the short rebuttal timeline. However, we agree that these would be interesting to explore, and we plan to include such experiments in the final version of our paper.
>
> For all experiments, we use SiT-B/2 as the base model and train for 400K steps. We observe that using DINOv2 outperforms the other self-supervised encoders. This result is consistent with the ablation conducted by  REPA.
>
>  **Model** | **Visual Representation** | **#PC** | **FID** |
> |:---------------|:--: |:--: |:----------------:|
> |   SiT-B/2            | -|- |           33.0 |
> |   SiT-B/2 w/ ReDi    |  MAE    | 8|            28.5 |
> |   SiT-B/2 w/ ReDi    |  MAE    | 12|          25.4|
> |   SiT-B/2 w/ ReDi    |  JEPA    | 8|            29.3 |
> |   SiT-B/2 w/ ReDi    |  JEPA    | 12|          25.7|
> |   SiT-B/2 w/ ReDi    |  DINOv2    | 8|         21.4 |
>
>
>
>
> **W3. A more in-depth explanation regarding Figure 7 (Principal Components Ablation)**
>
> We agree with the reviewer that further exploration is needed to better explain the performance degradation observed when increasing the number of DINOv2 PC's. To support our claim that this degradation occurs due to the model disproportionately allocating capacity to predict the visual representations at the expense of image latents, we conduct two additional experiments.
>
> One experiment with PC=16 and $\lambda_z=0.5$ and additionally an experiment where we use the 8 components *after the first 8 PCs* i.e. $z[8:16]$. We highlight these new experiments with **bold** in the table below:
>
>  **Model** | **PC** | **$\lambda_z$** | **FID** |
> |:---------------|:--: |:--: |:----------------:|
> |   SiT-B/2            | -|- |           33.0 |
> |   SiT-B/2 w/ ReDi    |  0:8    | 1|            25.7 |
> |   SiT-B/2 w/ ReDi    |  0:16    | 1|          29.1|
> |   **SiT-B/2 w/ ReDi**    |  **0:16**    | **0.5**|            **31.5** |
> |   **SiT-B/2 w/ ReDi**    |  **8:16**    | **1**|            **25.5** |
>
> These results show that although the 8 components after the first 8 PCs ($z[8:16]$) contain useful information and match the performance of the first 8 PCs ($z[0:8]$), modeling both together negatively impacts performance. This holds true *even when we adjust the loss weight to $\lambda_z=0.5$*.  These experiments support our explanation for the performance degradation observed as the number of DINO channels increases.

---

> > ### Comment · Reviewer_8DbC · 2025-08-04
> > **Response to Author**
> >
> > Thank you very much to the author for their efforts in responding. I believe all questions have been resolved, so I will maintain the score.

---

### Official Review · Reviewer_wr8D · 2025-06-22

**Clarity:** 3
**Significance:** 3
**Originality:** 3
**Rating:** 5
**Confidence:** 3

**Summary:**

The paper proposes to jointly model the image latents and semantic features (e.g., DINO) within the same diffusion process. During inference, it introduces representation-guided sampling to leverage the semantic features. Experimental results show that the proposed ReDi improves generative quality for both DiT and SiT, and achieves faster convergence during training.

**Questions:**

- How about the speed? The sampling speed might be slower than the baselines due to the extra prediction of the DINO feature?

**Ethical Concerns:**

["NO or VERY MINOR ethics concerns only"]

**Final Justification:**

After reading the rebuttal, I would like to raise my score to accept as the authors addressed my main concerns. Though some additional results were promised but not included in the rebuttal, the provided materials convinces me that the paper could make a solid contribution to the community.

**Limitations:**

Yes, limitation of impact discussed in Appendix.

**Paper Formatting Concerns:**

No formatting concern

**Quality:**

3

**Strengths And Weaknesses:**

## **Strengths**
- The motivation of the proposed method is well-explained. In the introduction it discusses the gap in discriminative capabilities between LDMs and representation learning methods.
- Compared to the existing work with similar motivation  e.g., REPA that achieves the semantic alignment via distillation, the paper proposes a new direction that encourages diffusion models to explicitly learn the joint distribution of the VAE latents and semantic features. It could offer new insights to the research community.
- The experiment results show the superiority of the proposed method

## **Weaknesses**
- There is no analysis of the generated DINO features. For example, it would be good to demonstrate their discriminative capabilities via linear probing results. In Figure 5, there are qualitative examples of the generated DINO features; however, it would be better to compare them with the ground truth DINO features as well.
- There is no experimental result with higher resolution (512 x 512) input. It would be good to show the proposed method can also effectively synthesize the semantic features and improve the generative quality of higher resolution inputs.

---

> ### Author Rebuttal · Authors · 2025-07-31
>
> We appreciate your insightful comments and efforts in reviewing our manuscript. Below, we provide our responses to each of your comments:
>
> **W1.1 Quantitative analysis of the generated DINO features**
>
> We appreciate the reviewer’s suggestion to demonstrate the discriminative capabilities of ReDi's generated features. Although the primary aim of this work is to improve generative image modeling, we agree that this is an interesting experiment.
>
> Since we operate on only the first 8 Principal Components (PC) of DINOv2 features, both ground-truth and generated representations have reduced discriminative capabilities, making linear probing less effective. Instead, we use Attentive Probing [1], which is more suitable for this low-dimensional setting.
>
> Our evaluation setup is the following: We first train an Attentive Probing Head on 8-PC DINOv2 features from ImageNet. Then compare the classification accuracy on 50K (50 for each ImageNet class) generated samples between:
> - **Ground-Truth Features:** PCA-reduced features extracted from generated images with DINOv2.
> - **Generated Features:** Features directly generated by ReDi.
>
> We employ SiT-XL w/ ReDi trained for 4M iterations and use Classifier-Free Guidance as in Table 2, to generate the samples.
>
>  **Features** | **Accuracy** |
> |:---------------|:----------------:|
> |   **Ground-Truth Features**            |           90.5 |
> |   **Generated Features**            |           81.2 |
>
>
> We note that the top-1 accuracy is high for both the ground truth and the generated features, since they are produced in a class-conditioned setting with CFG. This pushes the generated samples to align with the condition, thereby limiting diversity. As such, these results are not directly comparable to those typically reported in representation learning papers, such as DINOv2.
>
> The results demonstrate that ReDi’s generated features largely preserve the discriminative capacity of the ground truth DINOv2 features extracted from generated images.
>
> **W1.2 Visual comparison between ground truth and generated DINO features**
>
> We appreciate the reviewer’s suggestion to include a visual comparison between ground truth and generated DINO features. Because the NeurIPS 2025 rebuttal guidelines do not allow new figures to be uploaded at this stage, we cannot include the requested visualization in the rebuttal. We will include the suggested visual comparison in the final version of our paper.
>
> **W2. Higher Resolution (512 x 512)**
>
> We agree with the reviewer that it would be interesting to validate the effectiveness of ReDi in improving the generative quality of higher-resolution images. However, due to our limited computational resources and the limited time frame of the rebuttal period, we were unable to conduct an experiment on ImageNet512. However, we plan to include such experiments in the final version of our paper.
>
>
> **Q. How about the speed?**
>
> We respectfully emphasize that we have explicitly measured the sampling speed of ReDi in terms of throughput (generated images per second) and reported the results in Table 5. The Merged Tokens strategy (our standard approach), which preserves the original sequence length, achieves a throughput of 4.51, while Vanilla DiT-B/2 achieves 4.52. This demonstrates that ReDi (MR) maintains the same computational cost (99.78%) as Vanilla DiT-B/2.
>
> **References**
>
> [1] Scalable pre-training of large autoregressive image models. In ICML'24

---

> > ### Comment · Reviewer_wr8D · 2025-08-03
> >
> > Thank the authors for the thorough rebuttal. The quantitative results of the generated DINO features are convincing.  All of my concerns have been addressed.

---

### Official Review · Reviewer_NELV · 2025-07-02

**Clarity:** 4
**Significance:** 4
**Originality:** 3
**Rating:** 5
**Confidence:** 4

**Summary:**

- This paper presents a joint diffusion training approach using low-level VAE representations and high-level DINOv2 representations, called **ReDi**.
- The joint training significantly improves the resulting diffusion model across the board—for both conditional and unconditional generation. The results are in the same vein as REPA (Yu 2024), but ReDi achieves even greater improvements. Additionally, joint training speeds up convergence substantially.
- The paper also introduces a sampling-time self-guidance trick called **Representation Guidance**, where the guidance signal comes from the denoising high-level DINOv2 representation. This signal is amplified in a way similar to classifier-free guidance (CFG), and the method effectively narrows the performance gap between conditional and unconditional generation.

**Questions:**

- I’m personally curious about *why and how* jointly modeling “more stuff” (like DINO, which should make the task harder) actually leads to **faster convergence**. More broadly, I’d like to understand how joint modeling in general helps convergence.
    - To answer these questions, it would be useful to test different joint representation choices. For example, one potential baseline is to jointly model with the **class label** itself.
        - My hypothesis is that class labels might outperform DINO representations (though I could be wrong). If they do, it might suggest that the advantage comes from the sharpness of the conditioning signal.
        - This would also shed light on why Representation Guidance helps: it could serve as a “sharpened” signal that accelerates denoising, akin to conditional generation. While it doesn’t provide a strong condition from the beginning, it offers a slightly stronger guide during sampling, enhancing overall generation quality.

**Ethical Concerns:**

["NO or VERY MINOR ethics concerns only"]

**Final Justification:**

I think this is a strong paper with solid results and good experimental design. The gains are meaningful (especially the faster convergence and improved FID), and the idea of Representation Guidance is interesting and opens up a lot of directions.

I had some questions about why joint modeling helps and how RG works, and while those are still open to some extent, the authors’ rebuttal and extra experiments helped clarify things. I found the responses reasonable and convincing overall.

There’s definitely more to explore here, but the current version already stands on solid ground. I’m keeping my score at 5: Accept.

**Limitations:**

Yes.

**Quality:**

3

**Strengths And Weaknesses:**

**Strengths:**

- The improved convergence speed and convergence quality have practical significance. Achieving **23× faster convergence** and **cutting FID in half** is a very strong result—this alone is worthy of publication.
- In addition to the substantial FID improvement, the idea of Representation Guidance raises intriguing research questions (see Weaknesses).

**Weaknesses:**

- There is limited investigation into *why and how* joint modeling with DINO representations leads to improvements. Although REPA observed that aligning a diffusion model’s representation with DINO helps, a deeper mechanistic understanding is lacking. ReDi’s approach is also structurally different from REPA, which warrants a fresh explanation.
    - Relatedly, Figure 7 shows that increasing the number of DINO channels can degrade performance. The paper suggests this happens because the model devotes too much capacity to modeling the DINO representation. While plausible, the experiments may not be well-controlled enough to support that conclusion. For instance, the loss weighting should be adjusted to maintain a constant ratio between the VAE and DINO losses. Without this control, it’s difficult to draw strong conclusions.
    - Even with such control, an open question remains: different DINO channels may encode fundamentally different types of information, and perhaps only some of that information is beneficial for diffusion. The paper too quickly attributes the observed degradation to limited capacity, without adequately exploring whether the nature—not just the volume—of the DINO information plays a role.
- There is also limited analysis of why **Representation Guidance** (a form of self-guidance) is effective. It’s curious that a signal generated by the model itself—when amplified—can lead to better diffusion outcomes. Understanding this phenomenon might uncover important new insights.

---

> ### Author Rebuttal · Authors · 2025-07-31
>
> We sincerely appreciate the reviewer's thoughtful questions about the fundamental mechanisms behind our approach's success. While these open-ended inquiries deserve careful consideration, providing comprehensive answers within the rebuttal period is difficult. For now, we share below some additional experiments, our current perspectives and note that these questions inspire valuable future research directions - particularly a deeper theoretical or mechanistic analysis of the ReDi's joint diffusion mechanisms.
>
> **W1.1 Why and how joint modeling leads to improvements.**
>
> We agree with the reviewer's assessment that the significant performance improvement demonstrated by ReDi raises interesting research questions regarding the underlying mechanisms of our method.
>
> Both our work and REPA are motivated by the observation that diffusion models tend to learn features that lack the depth and versatility seen in self-supervised approaches like DINOv2. Our hypothesis is that this limitation arises because the denoising objective does not differentiate between meaningful semantic information and irrelevant image details.
>
> REPA showed that aligning the DiT representations with DINOv2 can mitigate the deficiencies of the denoising objective, enabling the model to learn stronger representations. This resulted in significant improvements, indicating that learning high-level semantic representations is essential for generative modeling.
>
> This observation directly motivates our joint modeling approach. Rather than relying on distillation, ReDi simultaneously learns stronger semantic representations by learning to denoise the first principal components of  DINOv2, which encode the high-level semantic information of the image.
>
>
> **W1.2 Loss weight and #PC**
>
> We thank the reviewer for their suggestion to further explore a more optimal weighting between ReDI's  VAE and DINOv2 denoising losses. We agree that such an exploration is an interesting direction that we plan to do in future research. As an initial step in this direction, we conduct an experiment with PC=16 and $\lambda_z=0.5$. Additionally, we conduct an experiment where we use the 8 components *after the first 8 PCs* i.e. $z[8:16]$. We highlight these new experiments with **bold** in the table below:
>  **Model** | **PC** | **$\lambda_z$** | **FID** |
> |:---------------|:--: |:--: |:----------------:|
> |   SiT-B/2            | -|- |           33.0 |
> |   SiT-B/2 w/ ReDi    |  0:8    | 1|            25.7 |
> |   SiT-B/2 w/ ReDi    |  0:16    | 1|          29.1|
> |   **SiT-B/2 w/ ReDi**    |  **0:16**    | **0.5**|            **31.5** |
> |   **SiT-B/2 w/ ReDi**    |  **8:16**    | **1**|            **25.5** |
>
> These results show that although the 8 components after the first 8 PCs ($z[8:16]$) contain useful information and match the performance of the first 8 PCs ($z[0:8]$), modeling both together negatively impacts performance. This holds *even when we adjust the loss weight to $\lambda_z=0.5$*. While we agree that a deeper exploration is necessary, this preliminary result supports our explanation for the performance degradation observed as the number of DINO channels increases.
>
>
> **W1.3 Different DINO channels may encode fundamentally different types of information.**
>
> We once again thank the reviewer for their suggestion to explore *which* principal components (PCs) are most beneficial for ReDi. Indeed, different PCA components encode different types of information. Specifically, the first PCs capture low-frequency global structures, while the last PCs capture high-frequency details of DINOv2. Thus, we agree that determining whether low-frequency, high-frequency, or a combination of both is most beneficial for joint modeling is an interesting question.
>
> In this direction, we conducted two additional experiments (**in bold**) where we fixed the number of PCs to 8 and varied their position, with the 1st PC indexed at 0:
>  **Model** | **#PC** | **Position** | **FID** |
> |:---------------|:--: |:-- |:----------------:|
> |   SiT-B/2            | -|- |           33.0 |
> |   SiT-B/2 w/ ReDi    |  8    | 0:8 (first)|            25.7 |
> |   **SiT-B/2 w/ ReDi**    | **8**   | **760:768 (last)**|           **29.4**|
> |   **SiT-B/2 w/ ReDi**    |  **8**   | **384:392 (middle)**|           **30.1** |
>
> These results indicate that the low-frequency semantic information is the most beneficial for ReDi.
>
>
>
> **W2. Why Representation Guidance (a form of self-guidance) is effective?**
>
> This is indeed an interesting observation, our intuition is that  Representation Guidance (RG) can be viewed through the same framework as Classifier‑Free Guidance (CFG):
>
>  - In CFG $\nabla \log p(x_t|c) -\nabla \log p(x_t)$ acts as the gradient of an implicit classifier (amplifying the likelihood of $p(c | x_t)$), producing images that follow the condition $c$.
>  -  In RG $\nabla \log p(x_t,z_t) -\nabla \log p(x_t)$ acts as the gradient of an implicit feature extractor  (amplifying the likelihood of $p(z_t | x_t)$), producing images that follow the (noisy) semantic structure $z_t$ at each step $t$.
>
> We will highlight this connection more explicitly in the final version of the paper.
>
>
>
> **Q1. Why and how jointly modeling “more stuff” (like DINO, which should make the task harder) actually leads to faster convergence?**
>
> More generally, jointly modeling multiple modalities has been systematically shown to benefit image and video generative models, offering strong scalability properties. We discuss such methods in the *Multi-modal Generative Modeling* paragraph of our Related Work section.
>
> Specifically for ReDi, we have already analyzed our hypothesis for the observed performance gains in response to W1.1.
>
> **Q2. Would jointly modeling the class label outperform DINO?**
>
> We believe that the reviewer raises an interesting question. As an initial experiment, we attempted to encode the class label as a one-hot vector and jointly generate it as a separate token. However, we observed that this straightforward implementation is not successful. Specifically, the model was unable to learn the discrete "class label" distribution (potentially due to the denoising objective), and the training loss increased over time rather than converging. Still, we consider this an interesting direction, and further experimentation beyond the rebuttal period is needed.

---

> > ### Comment · Reviewer_NELV · 2025-08-04
> >
> > Thanks for the detailed response and for running the additional experiments — they were really helpful. The PCA component studies and the loss weight variations in particular add useful insight and show that there’s a lot more to understand here. Overall, I think this is a very interesting paper, and I’m excited to see future work that digs deeper into these ideas.
> >
> > A few follow-up thoughts on the points we discussed:
> >
> > **1. On Representation Guidance (RG) and the analogy to CFG**
> >
> > I see the intuition you're going for with comparing RG to classifier-free guidance, but I’m not sure the analogy fully holds. In CFG, the guidance comes from an external condition `c`, which contains actual extra information. In RG, the guidance is based on the model’s own prediction `z_t`, so it’s less obvious how that adds anything new beyond what's already in the current state `x_t`.
> >
> > This makes me wonder if RG is maybe closer in spirit to something like Auto-guidance (e.g., Karras et al.). Those methods also try to improve generation using internal signals rather than external ones. It might be interesting to explore that connection more.
> >
> >
> > **2. On faster convergence from joint modeling**
> >
> > This was more of a curiosity than a criticism — I don’t think it affects the strength of the paper.
> >
> > That said, I find that “DINO encourages stronger representation in DiT” explanation is not straightforward. It maybe true, but it's not clear that it connects strongly to this question. I think exploring to pin-point the connection: why "stronger representation" helps diffusion generation? Is an interesting direction.
> >
> > You mentioned prior multimodal work, but as far as I can tell, many of those examples (e.g., 4M-21) are in the fine-tuning or transfer setting. I also looked at CoDi but didn’t find anything that really sheds light on this. Again, not a problem for this paper — it’s just a genuinely interesting puzzle that might be worth exploring more.

---

### Official Review · Reviewer_UuCC · 2025-07-02

**Clarity:** 4
**Significance:** 2
**Originality:** 2
**Rating:** 5
**Confidence:** 5

**Summary:**

This paper proposes a new technique, called ReDi, to improve the training and inference of generative diffusion models for images.
The paper places itself in the context of transformer latent diffusion.
The main idea of the paper is to learn to noise/denoise both the image latent and a secondary image representation derived from a pre-trained DINOv2 model.
The secondary image representation acts as regularizer like is commonly done in self-supervised learning.
Furthermore this secondary objective yields an additional signal for guidance during inference which the authors coin adequately as representation guidance.
Experiments on ImageNet256 demonstrate an improvement over REPA, in particular when NOT using Classifier-Free Guidance (CFG).
When using CFG, there's little room for improvement in the first place for ImageNet256, but the proposed method still manages to show a little improvement for class-conditional generation and a significant one for unconditional generation.

**Questions:**

I do not have questions, my only suggestion would to dig deeper on trying various secondary representations to help uncover what a good representation should have.

**Ethical Concerns:**

["NO or VERY MINOR ethics concerns only"]

**Final Justification:**

I am raising my score significantly as the points I raised were very satisfactorily answered, changing my initial perception of the paper.

**Limitations:**

I didn't see any discussion on limitations or potential negative social impact.
I would assume that it shares those with generative modeling in general, e.g. that ultimately could allow to generate fake data that is indistinguishable from reality.

**Paper Formatting Concerns:**

I didn't notice any issue.

**Quality:**

3

**Strengths And Weaknesses:**

Strengths
1. The paper is easy to read and clear
2. The method looks simple to implement
3. The experiments show convincing gains over REPA
4. Conceptually, I can see this applying to other domains than images but somehow this does require a (good) secondary representation to be available for the target domain.

Weaknesses
1. Experimental setting limited to ImageNet256
2. Secondary representation limited to one choice (DINOv2), it would have been interesting to try several alternatives for secondary representations.
3. Strength (4) raises the question of what is a good secondary representation, which in turn points back to weakness (2), this discussion is lacking in the paper
4. Consequently, the paper feels more like an engineering a solution to natural image generation that probing the underlying workings of diffusion models.

---

> ### Author Rebuttal · Authors · 2025-07-31
>
> We appreciate your insightful comments and efforts in reviewing our manuscript. Below, we provide our responses to each of your comments:
>
> **"A little improvement for class-conditional generation"**
>
> We respectfully clarify that ReDi's improvements on class conditional generation *are significant both with and without CFG*, even though we agree with the reviewer's assessment that "*there's little room for improvement in the first place for ImageNet256*" in the CFG setting. We summarize the performance gains over the baseline (SiT) and REPA in what follows:
>
> As presented in Table 2 ReDi reaches an FID of 1.72 at 350 epochs, surpassing both the baseline SiT and REPA trained for 1400 and 800 epochs, respectively. These results show that even in the CFG setting, ReDi accelerates convergence by **> x4** over SiT and by **> x2** over REPA.
>
>
> **W1. Experimental setting limited to ImageNet256**
>
> We appreciate the reviewer’s concern regarding the applicability of our method to other datasets.  To address this, we conducted an additional experiment on Food101 dataset [1]. While this serves as a preliminary result in a smaller-scale setting, it provides valuable empirical evidence that ReDi’s significant improvements are not limited to ImageNet256.
>
>
> | **Model** | **Iter** | **FID** |
> |:---------------|:--: |:----------------:|
> |     SiT-B/2   | 400K|           14.5 |
> |         SiT-B/2 w/ ReDi   |400K |            9.2 |
>
>
> Due to our limited computational resources and the short timeframe of the rebuttal period, we were unable to conduct a large-scale experiment, e.g., on ImageNet512. However, we plan to include such experiments in the final version of our paper.
>
>
> **W2-W3. Alternative representations**
>
> We thank the reviewer for their suggestion to explore alternative representations.
> Although we selected DINOv2 for its state‑of‑the‑art performance across a wide spectrum of vision tasks, we agree that evaluating other representations can provide additional insight. As such, we have already highlighted the potential for evaluating a combination of different representations in the "Limitations and Future Work" section (Appendix D).
>
> To address this concern, we conducted some preliminary supplementary experiments using 2 alternative representations, namely MAE and JEPA, with different number of principal components (#PC). Given the time constraints of the rebuttal period, a more thorough exploration was not possible. However, we plan to include additional alternative representations, such as CLIP, SIGLIP, and MOCOv3, in the final version of our paper.
>
> For all experiments, we use SiT-B/2 as the base model and train for 400K steps. We observe that using DINOv2 outperforms the other self-supervised encoders. This result is consistent with the ablation conducted by  REPA [2].
>
>  **Model** | **Visual Representation** | **#PC** | **FID** |
> |:---------------|:--: |:--: |:----------------:|
> |   SiT-B/2            | -|- |           33.0 |
> |   SiT-B/2 w/ ReDi    |  MAE    | 8|            28.5 |
> |   SiT-B/2 w/ ReDi    |  MAE    | 12|          25.4|
> |   SiT-B/2 w/ ReDi    |  JEPA    | 8|            29.3 |
> |   SiT-B/2 w/ ReDi    |  JEPA    | 12|          25.7|
> |   SiT-B/2 w/ ReDi    |  DINOv2    | 8|         21.4 |
>
>
>
> **W4. Engineering a solution**
>
> We respectfully disagree with the reviewer's assessment of our method as "an engineering solution for natural image generation".
>
>
> Modeling the joint distribution of precise low-level features (via the VAE latents) and semantic high-level features (via DINOv2) is well motivated by the recent literature [2] that established a connection between representation learning and generative performance.
>
> Further, we emphasize that *we did not apply any engineering tricks* during training (e.g., adjusting LR, optimization parameters, etc.) nor any changes in the standard diffusion transformer architecture (e.g., RMSNorm [3], SwiGLU [4], and RoPE [5]). Thus, our work presents a fundamental finding and not an engineering solution: integrating image latents and semantic representations within one probabilistic model demonstrably simplifies and improves diffusion‑based generation.
>
> **References**
>
> [1]  Food-101–mining discriminative components with random forests. In
> ECCV, 2014
>
> [2] Representation alignment for generation: Training diffusion transformers is easier than you think. In ICLR, 2025
>
> [3] Root mean square layer normalization. In NeuRIPS 2019
>
> [4] Glu variants improve transformer. In Arxiv 2020
>
> [5] Roformer: Enhanced transformer with rotary position embedding. Neurocomputing 2024

---

> > ### Comment · Reviewer_UuCC · 2025-08-04
> >
> > I am numbering my response for ease of discussion.
> >
> > > A little improvement for class-conditional generation
> >
> > R1: You are absolutely right, the convergence gains are massive and I overlooked that. I stand corrected.
> >
> > R2: Thanks you for doing some additional experiments, the results are exciting and I could definitely see future works probing what constitutes good alternative representations.
> >
> > > W4. Engineering a solution
> >
> > My initial assessment came from the limited choices of alternative representations, which translated a feeling of engineering a single solution that works (exploitation focus). In light of your new results with alternative representations, this addressed my original feeling, giving a more satisfying explorative flavor to the paper (I expect you to add these in the final manuscript).
> >
> > In light of the rebuttal feedback, I'm happy to substantially increase my rating.

---

### Decision · Program_Chairs · 2025-09-17

**Decision:**

Accept (spotlight)

**Comment:**

This paper enhances generative modeling by jointly modeling image latents and high-level DINO features, which significantly improves generative quality and training efficiency. The idea is well-motivated, the overall method is technically solid, and the results demonstrate clear improvements in faster convergence and reduced FID. Reviewers found the experimental design convincing, providing sufficient evidence of the method's impact and reproducibility. With unanimous support from the reviewers, the consensus is to accept the paper. AC agrees with this consensus and recommends accepting the paper.